# Syntactic Gender Agreement Processing on Direct-Object Clitics by Spanish-Speaking Children with Developmental Language Disorder: Evidence from ERP

**DOI:** 10.3390/children8030175

**Published:** 2021-02-25

**Authors:** Paloma Roa-Rojas, John Grinstead, Juan Silva-Pereyra, Thalía Fernández, Mario Rodríguez-Camacho

**Affiliations:** 1Dirección de Investigación, Instituto Nacional de Geriatría, Ciudad de Mexico 10200, Mexico; paloma_roa@hotmail.com; 2Department of Spanish & Portuguese, The Ohio State University, Columbus, OH 43210, USA; 3Proyecto de Neurociencias, Unidad de Investigación Interdisciplinaria en Ciencias de la Salud y Educación, Facultad de Estudios Superiores Iztacala, Universidad Nacional Autónoma de México, Tlalnepantla 54090, Mexico; marcizta@gmail.com; 4Laboratorio de Psicofisiología, Instituto de Neurobiología, Universidad Nacional Autónoma de México, Juriquilla, Querétaro 76230, Mexico; thaliafh@yahoo.com.mx

**Keywords:** specific language impairment, clitics, ERP, gender agreement, Spanish

## Abstract

Children with developmental language disorder (DLD) have a psycholinguistic profile evincing multiple syntactic processing impairments. Spanish-speaking children with DLD struggle with gender agreement on clitics; however, the existing evidence comes from offline, elicitation tasks. In the current study, we sought to determine whether converging evidence of this deficit can be found. In particular, we use the real-time processing technique of event-related brain potentials (ERP) with direct-object clitic pronouns in Spanish-speaking children with DLD. Our participants include 15 six-year-old Mexican Spanish-speaking children with DLD and 19 typically developing, age-matched (TD) children. Auditory sentences that matched or did not match the gender features of antecedents represented in pictures were employed as stimuli in a visual–auditory gender agreement task. Gender-agreement violations were associated with an enhanced anterior negativity between 250 and 500 ms post-target onset in the TD children group. In contrast, children with DLD showed no such effect. This absence of the left anterior negativity (LAN) effect suggests weaker lexical representation of morphosyntactic gender features and/or non-adult-like morphosyntactic gender feature checking for the DLD children. We discuss the relevance of these findings for theoretical accounts of DLD. Our findings may contribute to a better understanding of syntactic agreement processing and language disorders.

## 1. Introduction

### 1.1. The Phenomena

The development of direct-object clitics in Spanish and other languages has been the subject of much study. This stems from the fact that typically developing Spanish-speaking children pass through a stage during which they neither produce them in contexts where adults obligatorily would, nor do they mark them with gender and number agreement corresponding to their antecedents, as adults obligatorily would [1,2,3,4]. In this study, we will be most interested in non-adult-like agreement in children diagnosed with developmental language disorder (DLD); however, we will briefly describe both object omission and agreement errors in typically developing children. In this way, we aim to distinguish the kind of error we are interested in from the kind that is less directly relevant to our theoretical claims.

#### 1.1.1. Null Objects

Typically developing children across a range of languages show evidence of using direct objects of the verb in non-adult-like ways. One way in which their language is non-adult-like is their failure to produce a direct object of a transitive verb (where the adult grammar would require it), as in the following examples.

Spanish (Simon-Cerejido and Gutiérrez-Clellen [5] p. 336)

Child: El niño agarró.

the boy grabbed

“The boy grabbed.”

2.Bulgarian (Ivanov [6] p. 196)

Child: Ritna.

kicked

“He kicked.”

3.Greek (Marinis [7], p. 11)

Child: Aniki Ula.

open Ula

“Ula shall open.”

Sometimes these object omissions appear to take place in contexts that would require a pronominal in the adult language, either a clitic pronoun, or a tonic, free morpheme pronoun, according to the language. Other times, the omission occurs in a context in which pronominalization would not seem to be called for. The pronominal contexts are those in which the antecedent is made prominent in the immediately preceding discourse, while the non-pronominal contexts are those that lack such an antecedent. Each is illustrated in the following French examples from Pirvulescu and Roberge [8] from the Champaud Corpus of the CHILDES (Child Language Data Exchange System) Data Base [9].

4.Pronominal Null Object Context (Grégoire, 1; 11.22)

Adult: La pièce elle est dedans, oui.

the coin it is inside yes

“The coin it is inside, yes.”

Child: Enlever.

take out

“Take out.”

5.Non-pronominal Null Object Context (Grégoire, 2; 5.13)

Child: Remonte tout seul (he is trying to pull up his pants).

pull up all alone

“Pull up myself.”

This generalization is true in languages that require referential objects to be overt, such as Spanish (Here we refer to varieties of Spanish that do not allow referential null objects, though other varieties of Spanish, including at least those that are in contact with Euskera [10], Guaraní [11] and Quechua [12], do allow such null objects. See Schwenter [13] for a review), and it is true in languages in which referential null objects are a legitimate grammatical form in the adult language, including Portuguese [14] and Mandarin [15]. That is, children seem to use even more null objects in child Portuguese and child Mandarin than do adults.

In their study of direct-object omission, Castilla and Pérez-Leroux [1] showed in a sample of 103 monolingual Spanish-speaking children in Colombia that object omission occurs in 25% of elicitations by 3-year-olds, 15% in 4-year-olds and 13% in 5-year-olds (p. 14). Though other studies, using elicited production and other methodologies, find different percentages (cf. [2,3]), at least in this large monolingual sample, using a standard elicited production methodology, object omission seems to be well attested in child Spanish.

Though there is an active debate regarding the cause of non-adult-like object omission in child language, perhaps the most compelling evidence for lexical development as the principal cause comes from Pérez-Leroux et al. [4], who showed, using a structural equation model (SEM), that lexical development predicts object drop, while it does not, for example, predict determiner drop. Determiners, unlike direct objects, do not appear as a function of verb-specific lexical properties (i.e., transitive vs. intransitive), but rather are fully morphosyntactically productive. This predictive relationship of lexicon on object drop supports their claim that children have not yet learned which verbs are obligatorily transitive (e.g., devour), which are optionally transitive (e.g., eat) and which are obligatorily intransitive (e.g., laugh). This kind of information is, after all, lexically specific and would have to be learned on an item-by-item basis. In contrast, such is not the case for definite determiners, which are used as a function of regular, productive morphosyntax across the board. We will not have anything more to say about this debate in the remainder of this article, but rather limit ourselves to the simple observation that lexical development would seem to be a conceptually critical component of the development of this aspect of child language.

#### 1.1.2. Agreement Errors

Failing to produce direct objects of verbs is not the only non-adult-like dimension of this phenomenon, however. When children attempt to produce direct-object clitic pronouns, they often produce errors of gender and number agreement with the 3rd person antecedent. In Castilla and Pérez-Leroux’s [1] (p. 14) typically developing monolingual sample, gender errors were produced in 3% of elicitations with 3-year-olds, 2% with 4-year-olds and 4% with 5-year-olds. Similarly, with number errors, there were 11% number errors with 3-year-olds, 8% with 4-year-olds and 5% with 5-year-olds.

Critically, Castilla and Pérez-Leroux [1] reported that object omissions and agreement errors did not correlate in their sample, which is consistent with the idea that they arise from different causes (While some studies concern themselves with person agreement errors, also, we will limit ourselves, following Castilla et al. [1] and others, in focusing on 3rd person clitics). Among the possible errors that children can make with direct-object clitics, then, gender agreement errors between the direct-object clitic pronoun and its antecedent are what we will be concerned with in this study.

### 1.2. Interface Delay, Interface Deficit and Definites

What is theoretically interesting about agreement errors with object clitics in the developing language of monolingual Spanish-speaking children? Does this kind of error fit into a larger pattern of errors produced by children in general and children with DLD? To answer these questions, it could help to think about the type of construction that we are considering. Notice that direct-object pronominal clitics refer, via anaphora or deixis, to an antecedent that is prominent in the preceding discourse or that is made prominent via ostension in the physical context. In the following utterance, we see an anaphoric context. An indefinite Determiner Phrase (DP) is presented in subject position of the first sentence, and is followed in the second sentence as a direct-object clitic (“la”), which agrees in person, number and gender with the antecedent, third person, singular, feminine “una niña”.

6.Anaphoric Context

Una niña de nuestro equipo marcó un gol. La debes felicitar.

a girl of our team scored a goal. pronoun 3rd sg. should-2nd sg. pres. congratulate-inf.

“A girl from our team scored a goal. You should congratulate her.”

In the following deictic context, a waiter brings freshly foraged mushrooms to a table to show the customers. The chef is going to make them into ravioli, but one of the customers is very hungry and says the following.

7.Deictic Context

Los quiero comer ya.

pro-acc. want-1st sg. pres. eat-inf. already

“I want to eat them right now.”

In this physical context, the pronoun “los” agrees in person, number and gender with the third person, singular, masculine noun “hongos” or mushrooms. Given the physical context, the speaker could either point or otherwise gesture towards the mushrooms or discourse pragmatics might allow the “unheralded” use of the pronoun, presupposing the interlocutor is as familiar with the unspoken referent of the pronoun as the speaker is.

In both cases, the referents of the pronouns have become prominent in either the previous discourse or by inference from the physical context, what Stalnaker [16] referred to as the Conversational Common Ground. Roberts [17,18] argued convincingly that both such pronouns should be considered definite expressions, inasmuch as they are licensed in contexts in which the speaker presupposes that interlocutors are familiar with them and that they are unique in context, in the same way that noun phrases modified by definite articles are. Following this account, other definites would include tonic pronouns (e.g., él—he, ustedes—you-pl, etc.), names (e.g., Ramón, Melissa, Ximena, etc.), definite noun phrases (e.g., el auto tuyo—your car) and null subjects (e.g., Ya llegó Juan. *Ø* se había ido hace una hora.—John just arrived. (He) had left an hour ago.). This semantic natural class of definites is interesting in that they, as a class, seem to develop later in child language than do constructions that do not require access to the Conversational Common Ground.

There is a sense in which the Optional Infinitive verb phenomenon (e.g., [19]) forms part of this natural class, as well, inasmuch as verbal anaphora, in the sense of Bittner [20], denotes a relationship between speech-time and event-time, following Reichenbach [21], that is taken to be familiar to both speakers and interlocutors. Thus, both nominal and verbal anaphora take time to develop in typically developing children. We contrast these constructions with more syntactically local relationships, such as nominal plural marking, noun-determiner agreement or noun–adjective agreement, which seem to develop relatively earlier in child language than do definites (e.g., [22,23,24]).

Given this pattern, one might be tempted to hypothesize that it is human discourse-pragmatic abilities, as instantiated, for example, in the components of Theory of Mind [25,26] that develop, and not syntax itself. More than one insightful cognitive scientist has in fact made just such a proposal (e.g., [27,28]). However, there is also evidence that multiple components of Theory of Mind are quite adult-like at an early age, even in infancy. Belief tracking—one Theory of Mind component—seems well-developed in 15-month-old pre-linguistic infants [29]. Furthermore, work on intention tracking, another subcomponent of the Theory of Mind construct, in Woodward et al. [30], shows that 12-month-old infants appear able to track the intentions of those around them. Similarly, outside the domain of Theory of Mind, but still in the domain of discourse pragmatics, Baker and Greenfield [31] gave evidence that 2-year-old children have knowledge of new versus old information in spontaneous production, before adult-like morphosyntax was being used in their English. Taken together, what is known about plausible non-linguistic cognitive abilities that could constitute important dimensions of discourse-pragmatic knowledge suggests that this knowledge, to the degree that we can separate it from language, appears well-developed in infants, who are not yet able to use definites.

On this basis, it would seem ill advised to conclude that definites are slow to develop in typically developing children as a function of discourse pragmatics itself being slow to develop. Given the fact that so much else in morphosyntax has developed to relatively adult-like levels (e.g., declarative and interrogative word orders, plural marking, preposition-object word order, etc.) at the point at which definites are still a struggle for children, researchers have proposed the hypothesis they refer to as Interface Delay [32,33] for typically developing children and Interface Deficit [34,35] for children diagnosed with DLD. These hypotheses attempt to account for the difficulty of acquisition of these constructions in terms of the inability of linguistic and discourse-pragmatic domains of cognition to interact with one another. The relationship between these domains, assuming a modular cognitive architecture (e.g., [36,37,38]), appears to become more robust and facile gradually over the course of development. A perhaps useful analogy for this development could be the relationship between the two language systems of bilinguals, if they study to become interpreters. Any bilingual has both systems, which can more or less interact, but only interpreters go back and forth quickly and accurately between the two. Similarly, the domain of language may be well developed on its own and the domain of discourse pragmatics may be well developed on its own in the developing mind of a child, but to become a competent, neurotypical adult user of definites, both systems must work together in an agile, robust fashion.

### 1.3. The Unique Checking Constraint (UCC), the Computational Complexity Hypothesis (CCH), Interface Deficit and Clitic Agreement Errors

In previous work, Grinstead et al. [39] have argued that Interface Deficit can account for at least the linguistic phenomena that appear difficult for children with DLD. Another prominent proposal, the UCC (see [19,40,41]), has produced some excellent empirical work and raised the bar for linguistic study in the domain. It reports on both expressive and receptive dimensions of the tense deficit in a child English DLD sample, studied longitudinally. The theory-specific Minimalist [42] formulation of the UCC, however, would seem to over-exclude constructions that should be problematic in the grammars of Spanish-speaking children with DLD on the UCC account, such as noun–adjective agreement (e.g., dos gatos negros—two cats-masc. pl. black-masc. pl.), which do not in fact appear to be problematic in monolingual Spanish-speaking children in Mexico [43] (Though Bedore and Leonard [44,45] found different results with Spanish-speaking children in the US context). That is, the UCC claims that multiple features occurring in the derivation of a construction (number and gender in this case; tense and agreement in the original formulation designed to account for the Extended Optional Infinitive Stage in DLD) should be sufficient to make a construction subject to prolonged and severe difficulty in the language of children with DLD. However, this does not seem to be the case for noun–adjective agreement in child Spanish DLD. The same could be said of, for example, plural marking in Spanish, at least on Picallo’s [46] account of the adult morphosyntax of plurals, which, again, is not problematic for monolingual child Spanish-speakers in Mexico, diagnosed with DLD [43].

Noun plural marking in Spanish DLD would also seem to be predicted to be problematic by another prominent account of DLD, namely the Computational Complexity Hypothesis (CCH) of Jakubowicz and Nash [47], which claims that constructions will be difficult for children with DLD as a function of their relative necessity and as a function of whether they are required by syntactic versus semantic motivations. These criteria are designed to explain why the present tense in French does not seem to be difficult for children with DLD, that is, person morphology is used to mark present tense and is required in all tenses, and is therefore necessary, while past tense is more complex, in that it is required by semantics and not always necessary. One could argue that plural marking on nouns is required by semantics, but not always present, as on singular nouns, and thus should be classified as computationally complex by the CCH. On this basis, plural nouns should consequently be difficult for Spanish-speaking children with DLD, which, as we have discussed, does not appear to be the case in monolingual Spanish-speaking children in Mexico. It is for this reason that a linguistic account of DLD would seem to need to take into consideration the discourse-sensitive versus discourse-insensitive distinction, which neither the UCC nor the CCH do, but which Interface Deficit does.

Beyond these constructions that seem difficult for other frameworks to handle, there is a range of constructions that are problematic for children with DLD that directly illustrate the leading idea of Interface Deficit: anaphora is difficult. Definite noun phrases, for example, are problematic, according to work by Anderson and Souto [48] and Restrepo and Gutiérrez-Clellen [49], null subjects are problematic [50], as is tense marking in Spanish [35]. It has, of course, been demonstrated to be a highly specific and sensitive clinical marker of DLD in English (e.g., [51,52]) and has been shown to be problematic in other languages of children with DLD, including French [53], Dutch [54] and Hebrew [55]. Finally, and most relevant to our study here, direct-object clitic pronouns are difficult in monolingual child Spanish DLD, and agreement errors in particular, are prominent [3,56,57].

To address the construction that we propose to study here, how should agreement errors in clitic production be treated theoretically? The proposal in Wexler et al. [2], following the UCC, predicted that clitic errors should not occur in typically developing Spanish-speaker or in Spanish-speaking children with DLD, given that, on their account of the syntax of clitics, only 1 relevant feature is involved, which should exempt the construction from non-adult-like grammar. From the perspective of the CCH, one might predict that clitic agreement errors should occur in that they are not always present, given that not all verbs take direct-object arguments, and because they are required semantically to complete the theta grid of the transitive verb with which they occur. From the perspective of Interface Delay, direct-object clitics are definites, and consequently occur as a function of anaphora and should therefore be problematic. Furthermore, gender and number agreement, specifically, are going to depend on anaphora to a previously mentioned antecedent, making an interface between syntax and discourse pragmatics critical. Though our study of clitic agreement errors will not adjudicate between Interface Delay and the CCH, as they appear to make identical predictions, the UCC does appear to predict that they should not be problematic. We note, for completeness, that a distinct interpretation of the UCC, in which gender and number were taken to be the relevant features, and not participial agreement generally, could make the UCC fall in line predictively with the CCH and Interface Deficit.

### 1.4. The Surface Hypothesis and Clitic Agreement in DLD

Finally, another popular account of the DLD deficit is the Surface Hypothesis of Leonard [58], inter alia, which is explored in Aguilar-Mediavilla, Sanz-Torrent and Serra-Raventós [59] as an account of function word and weak syllable omission in bilingual Spanish-Catalan-speaking children. The core claim of the Surface Hypothesis is that children struggle to perceive words and morphemes with low phonetic salience, which results in them creating grammatical systems that do not incorporate these elements or incorporate unstable or otherwise defective versions of them. The intervening years since the Surface Hypothesis was proposed, however, have produced a number of strong counter-arguments to this core claim. First, and perhaps most compellingly, child English-speakers diagnosed with DLD produce plural /s/ with high levels (83% correct) of accuracy [60,61]. This is problematic for the Surface Hypothesis inasmuch as it predicts across-the-board difficulty with morphemes as a function of their phonological properties, which make plural /s/ in English identical to the /s/ that marks third person singular present tense. Rice and Wexler [62], convincingly, report that, in both elicited production and spontaneous production, child English-speakers diagnosed with DLD produced noun plural /s/ correctly 88% of the time, while third singular present tense /s/ was produced correctly 35% of the time, at best. This comparison is critical because “Three rocks.” and “He walks.” are segmentally and phonotactically identical. This contrast is mysterious on the Surface Hypothesis, but explicable following syntactic and semantic explanations, such as those just reviewed.

How would the Surface Hypothesis work in Spanish for clitic agreement? To begin with, plural marking is also not problematic in Spanish DLD, as we have already seen. This means that the general plausibility of a hypothesis about low phonetic salience driving the morphosyntactic deficit in Spanish is low. More to the point, the core question of our project addresses gender agreement on clitics. Grinstead et al. [43] reported that noun–adjective agreement, including gender agreement, is not significantly different between DLD and control groups on an elicited production task. While not the identical version of gender agreement, as they were not looking at clitics, these are still the same segmental “o” and “a” vowels that mark gender on both morpheme types. This finding in fact seems consistent with Aguilar-Mediavilla et al. [59], who did not report significant differences between their DLD and control groups for agreement and who also, like Castilla and Pérez-Leroux [1], found no correlation between omission and agreement errors. In sum, though it is a promising account of inflectional morphology in actual deaf and hard-of-hearing children (e.g., [63]), who show across-the-board deficits, including both tense marking and plural marking in English, the Surface Hypothesis does not appear to be an adequate account of the DLD deficit.

### 1.5. Clitic Agreement Errors in Spanish DLD

Previous work on the production of direct-object clitics in Spanish-speaking children with DLD, summarized in Table 1, has been done with both monolingual Spanish-speaking children [3,57,64,65] and bilingual Spanish-speaking children, whose other language was English [44,45,66,67,68] or Catalan [69]. In addition to monolingual versus bilingual status, these studies also varied methodologically in that some used elicited production or cloze-type tasks, while other used either Frog Story/story-retell tasks or less-structured spontaneous production tasks. Also, children varied by age between preschool and early school age. Common to all of them was the finding that children diagnosed with DLD made errors of number and gender agreement, which in some cases was at a higher rate than typically developing control groups and in other cases was not.

Summarizing the findings from previous work on the topic of number and gender agreement in direct-object clitics in the language of child Spanish-speakers diagnosed with DLD, it seems fair to say that agreement errors are ubiquitous and persistent. More specifically, we see that they occur at least some of the time in children in the 5-year-old range who are monolinguals, and that there are significant differences between children with DLD from typically developing age controls. These specific contrasts are critical to us, as these are the populations we will test in this study.

We now ask whether online measures, specifically, Event-Related Potentials, might also contribute to what is known, by providing either convergent or divergent evidence of a deficit with a type of definite construction, direct-object clitics.

### 1.6. Event-Related Potentials and Agreement in Children with DLD

Event-Related Potentials (ERPs) provide a measure of the brain electrical activity temporally associated with an event, which can be sensory, motor, or cognitive. ERPs can provide information about language processing with highly precise temporal resolution and are classified according to their polarity (i.e., positive or negative deflections in the waveform), the time of their peak occurrence in milliseconds, and their topographical distribution across the scalp [70]. ERP studies of sentences processing have analyzed anaphoric relationships using agreement features (i.e., person, number, and gender) to understand syntactic and semantic dimensions of the processing of pronouns, including agreement [71]. ERP studies of younger and older adults on Spanish sentence processing have shown that morphosyntactic agreement mismatches usually show left anterior negativities (LAN) as compared to agreement matches, which occur between 300 and 500 ms, followed by a positive wave that emerges between 500 and 1000 ms (i.e., P600) after stimulus presentation [72,73]. Functionally, a LAN is taken to reflect automatic morphosyntactic parsing [73,74,75], while the P600 is thought to represent processes of syntactic revision, including reanalysis or repair (e.g., [76,77]). Adult-like LAN and P600 effects have been found in normal children as young as 2 years old [78,79], but other studies have shown only a delayed P600 effect in normal toddlers when syntactic violations are presented to participants [80].

Weber-Fox et al. [81] studied subject–verb agreement processing in teenagers with DLD (e.g., Every day, the children *pretends/pretend to be super-heroes.). They observed a right anterior negativity (RAN) in both control and DLD groups, and a reduced P600 in the DLD group as compared to controls. In contrast, in another study where syntactic errors of word category in German were analyzed, whereas typically developmental (TD) children showed a bilateral early starting anterior negativity (ELAN) and a posterior P600, children with DLD showed a comparable P600 but, unlike the TD children, there was only a late, clearly left-lateralized anterior negativity [82].

Different results have been observed in the comparison between TD children with respect to DLD children using grammatical and ungrammatical questions (Who did Joe see someone?/Who did Joe see?/syntactic error) [83], because whereas TD children displayed ELAN effect for the processing of questions containing a syntactic error, children with DLD did not. Epstein et al. [84] studied whether children with DLD had atypical processing of subject and object wh-questions. Children with DLD comprehended both question types poorly, and they found a smaller sustained positivity effect in the children with DLD, compared to TD children.

According to these findings, we should expect that the electrophysiological brain response of Spanish-speaking children with DLD would be different from that of typically developing children, when exposed to gender mismatches between direct-object clitics and their antecedents. In particular, we might expect ERP waveforms to reveal a sustained LAN effect, as has been observed in previous ERP studies, with children of other ages. Further, LAN amplitudes would likely be smaller in children with DLD than in TD children. This expectation is based on evidence that the syntactic deficit, characteristic of DLD children, is correlated with a smaller LAN effect for processing of morphosyntactic errors, like gender mismatch.

### 1.7. Summary and Research Questions

In summary, the two most prominent non-adult-like phenomena involving direct-object clitic pronouns in child languages, including Spanish, are the fact that typically developing children and children diagnosed with DLD omit pronominal and full DP direct objects and that, when they do use them, there can be a failure of number and gender agreement between the clitic pronoun and its antecedent. Gender agreement errors, the focus of our study, do not correlate with object omission, in the one large-sample behavioral study so far conducted. This suggests that the object of our study is an independent phenomenon. Two of the prominent grammatical theories of DLD (Interface Deficit and the CCH) predict that gender agreement errors should occur in children with DLD, though for different reasons, while a third, the UCC, does not. The four existing studies of monolingual Spanish-speaking children with DLD coincide in showing with behavioral measures (elicited and spontaneous production) that this agreement is indeed a problem. No electrophysiological measures have thus far been used to compare typically developing children and children diagnosed with DLD on this dimension of grammar. The work that has been done using ERPs on subject–verb agreement in children suggests that a smaller LAN or P600 in children with DLD versus Typically developing children should be expected. If this is indeed what we find, it would serve as converging evidence of the behavioral generalization that children with DLD struggle with this construction.

In light of what we have considered, it would seem possible to distinguish children diagnosed with DLD from typically developing children using ERP components corresponding to antecedent-clitic pronoun gender agreement mismatches, which leads us to the following research questions:Do child Spanish-speakers show the LAN-type effect we expect for gender agreement errors?If so, is this effect less pronounced for children diagnosed with DLD than it is for typically developing children of the same age?

## 2. Materials and Methods

### 2.1. Participants

Thirty-five monolingual Spanish-speaking children in public schools from Mexico City participated in our experiment. The mean age for the group was 73 months (6 years, 1 month) with a standard deviation of 14.06 months. All children were given pure tone hearing tests and passed at conventional levels. Further, none of them had experienced recent episodes of otitis media. All children also scored above 85 on a test of nonverbal intelligence, the Test of Nonverbal Intelligence (TONI-2; [85]). This was done to meet the former, stricter definition of specific language impairment, which all of our TD and DLD children meet, in addition to also meeting the DLD definition of Bishop et al. [86]. The children had neither social nor physical impairments to communication, oral structural problems, or frank neurological damage, which was determined at an initial diagnostic examination and parental consultation. All children were given the Battery for Language Assessment ([Batería del Lenguaje Objetiva y Criterial Screening] BLOC-S; [87]), which has 118 items, divided into four main modules that measure distinct domains of linguistic knowledge (morphology, syntax, semantics, and pragmatics). This battery has been used in other studies for diagnosis and treatment of monolingual Spanish-speaking children with DLD (e.g., [88]).

The children who had scores above the 85th percentile on all BLOC-S subtests were assigned to the typically developing group. The children who had scores on at least two of the BLOC-S subtests that were less than 1.25 standard deviations below the mean were assigned to the DLD group. Furthermore, all children in the DLD group had received an educational diagnosis of language impairment by the psychologist of their schools. The mean age of the 16 children in the DLD group was 73.73 months (SD = 11.46), while the mean age of the 19 children in the typically developing control group was 72.45 months (SD = 16.01). The groups were not significantly different from one another in age (t(33) = 0.857, *p* = 0.398).

At the beginning of the first testing session, after an appropriate explanation, informed consent was obtained from all participants according to Helsinki Declaration guidelines. The protocol was approved by the Ethics Committee from the Institute of Neurobiology at the National Autonomous University of Mexico (UNAM). Parents or legal guardians also provided written consent.

### 2.2. Procedures

#### 2.2.1. Stimuli

Children were presented with audio-recorded sentences of the following declarative type. Each sentence contained a transitive verb, presented in the third person singular, preterit (past perfective):8.El papá lo filmó.

the father him filmed

“The father filmed him.”

These sentences were presented simultaneously with an image that either matched the gender of the 3rd person singular animate direct object of the verb, or did not match (e.g., lo vs. la—him vs. her). The sentence in 8, for example, would not match the image in Figure 1, as the gender of the child in the image is stereotypically feminine.

The direct object in all cases was either niña (girl), niño (boy), gata (cat-fem.), gato (cat-masc.), perra (dog-fem.) or perro (dog-masc.). The subjects varied among abuela (grandmother)*,* abuelo (grandfather)*,* gato (cat-masc.)*,* hermana (sister)*,* hermano (brother)*,* maestra (teacher-fem.)*,* maestro (teacher-masc.)*,* mamá (mother)*,* papá (father)*,* perro (dog-masc.)*,* señor (gentleman) or señora (lady). These subjects and objects were composed into sentences with 50 verbs selected from two corpora of child-directed speech in Mexican Spanish, to ensure that the vocabulary used would be age appropriate. The first comes from “La producción del lenguaje de niños mexicanos” [Language production of Mexican children]; [89] and Cómo usan los niños las palabras? [How do children use words?]; [90] and the second comes from “Spanish Screener for language impairment in children” (SSLIC; [91,92]).

Thus, each verb was paired with one of the 6 direct objects and one of the 12 subjects in such a way that 50 gender-congruent sentence–image pairs were created. Then each of these same 50 sentences was presented with the corresponding image, except that the gender of the direct object was switched, yielding 50 gender-incongruent sentence–image pairs, as in Figure 1. The target stimulus in all cases was the onset of the image-congruent or image-incongruent direct-object clitic. All sentences have the same number of words before and after the clitic (see Appendix A). There was no significant difference in duration between the congruent and incongruent sentences (*p* > 0.05). Each child listened to the same sentence twice, once in the agreement and once in the disagreement condition.

To ensure the time-lock between the clitic in the audio file and the ERP recording, the wave forms of all audio files were carefully inspected and marked at the onset of the clitic. Additionally, 25 image-congruent and image-incongruent sentences were used as filler items. Congruence in the filler sentences varied as a function of the theta role of the participants in the image (e.g., La niña la saludó. “The girl greeted her.” with an image in which a grandmother is greeting a girl.) All sentences were spoken by a female native speaker of Mexican Spanish and were recorded on a digital-audio system, sampled at 20 kHz with a 16-bit resolution in stereo. The speaker rehearsed all the sentences prior to recording them to ensure that they were produced fluently. The average sound pressure level ranged from 63 to 67 dB sound pressure level (SPL).

#### 2.2.2. Event-Related Potentials Recording

The EEG was recorded from 32 tin electrodes secured in an elastic cap (Electrocap, CompuMedics, Eaton, OH, USA) at the following locations (according to the international 10–20 system): Fp1, Fpz Fp2, F7, F3, Fz, F4, F8, FT7, FC3, FCz, FC4, FT8, T7, C3, Cz, C4, T8, TP7, CP3, CPz, CP4, TP8, P7, P3, Pz, P4, and P8. The electrooculogram (EOG) was also recorded from one supraorbital electrode and an electrode placed on a child’s left cheek. The recordings were referenced against the left mastoid, and brain electrical activity over the right mastoid was also recorded. Offline, all electrodes were re-referenced to average mastoids. Electrode impedances were kept below 15 kOhms. The EEG signal was amplified with Neuroscan amplifiers (CompuMedics, NeuroScan Inc., Charlotte, NC, USA). A 200-Hz sampling rate was used to digitize the EEG, with a band-pass filter set from 0.1 to 100 Hz.

The child was seated in a comfortable chair, 70 cm from the computer screen in a sound-attenuated dimly lit recording chamber. All children were instructed to relax and maintain their gaze towards the center of the screen and to avoid blinking. Subjects passively listened to the stimuli while watching each scenario. The pictures appeared in the center of the screen, thus decreasing eye movement artifacts. Each spoken sentence was presented via headphones which were placed on children’s heads. Each session began with a visual presentation of all the characters used in the pictures, which concluded once the child demonstrated that they were familiar with each one. The purpose of this step was to assure that children were familiar with the names and corresponding genders of the nouns with which they were presented, to support our measurement of their reactions to the gender mis-match. Each item began with the presentation of the visual stimuli for 800 ms and after 250 ms an auditory stimulus was presented for 550 ms. The inter-stimulus interval was 1500 ms.

The entire session, including fitting the cap, lasted 45 min to 1 h. Sentences were presented in a randomized order. Each participant received the following number of sentences per condition: 50 in the gender-matched condition, 50 in the gender-mismatched condition and 50 filler items.

### 2.3. Data Analysis

ERPs were computed off-line from 1100 ms epochs for each subject in the gender-matched and gender-mismatched conditions. Epochs were comprised of the 100 ms preceding and the 1000 ms following the presentation of the clitic. EEG epochs with electrical activity greater than +/−150 μV and amplifier blocking for 50 ms or more at any electrode site were considered artifacts and the whole segment was automatically rejected. EEG epochs with artifacts due to eye movements or excessive muscle activity were eliminated by visual inspection off-line before averaging.

Subjects with fewer than 50% artifact-free trials for each condition were excluded from the average. Five DLD subjects and 7 TD subjects were excluded at this step of the process, leaving 11 children with DLD (Mean age = 5;8, SD = 1.00) and 12 controls (Mean age = 5;9, SD = 0.71), whose data could be analyzed. The mean ages for the two groups were not significantly different (*p* > 0.05). Baseline correction was performed in relation to the 100 ms pre-stimulus time mentioned above.

#### Statistical Analysis

Statistical analyses were performed on mean amplitude values from two time windows, which could potentially have carried relevant ERP signatures for the gender match/mismatch comparison of interest: 250–450 ms and 500–850 ms. These time windows were determined according to previous agreement studies and from inspection of both individual and grand average waveforms. Repeated-measures ANOVAs were performed separately for each time window. Separate four-way ANOVAs were performed for each time window using Group (DLD vs. TD) as a between-subject factor and Agreement (match vs. mismatch), Anterior–posterior (Frontal [F7, F3, Fz, F4, F8], Frontal-central [FT7, FC3, FCz, FC4, FT8], Central [T7, C3, Cz, C4, T8], Central-parietal [TP7, CP3, CPz, CP4, TP8], Parietal [P7, P3, Pz, P4, and P8]), and Coronal (Left, Middle-left, Middle, Middle-right and Right) as within-subject factors. The Huynh–-Feldt epsilon was applied to the degrees of freedom of those analyses with more than one degree of freedom in the numerator. Tukey’s honest significant difference (HSD) post hoc tests were completed after the ANOVA.

## 3. Results

Grand average ERP waves elicited by the critical word (match vs. mismatch conditions) for both groups are shown in Figure 2. In the TD group, morphosyntactic gender violations elicited a negative shift starting at 250 ms (i.e., anterior negativity), and ending at around 500 ms. There was no such effect observed in children with DLD. A positive wave followed this LAN-like effect in the TD children.

### 3.1. ERP Time Window 250–500 ms

There was an important, significant Group by Agreement by Anterior-posterior interaction (F(4, 84) = 3.92, *p* = 0.034, epsilon _H-F_ = 0.430, η^2^_p_ = 0.157). Post Hoc analyses showed in TD children a LAN effect (i.e., greater amplitude of negativity in the mismatch than with the match condition) in frontal (MD_HSD_ = 3.22, *p* = 0.001) and frontal-central areas (MD_HSD_ = 2.36, *p* = 0.014). In contrast, children with DLD displayed no such effect in frontal (MD_HSD_ = 0.22, *p* = 0.81) and frontal-central areas (MD_HSD_ = 1.11, *p* = 0.24).

ANOVA results also revealed significant Group by Anterior-posterior by Coronal interaction (F (16, 336) = 3.89 *p* = 0.001, epsilon = 0.378, η^2^_p_ = 0.156). Post hoc comparisons indicate the occurrence of, for both agreement conditions, smaller amplitudes of negativities for TD children than there were for children with DLD in the right frontal area (MD_HSD_ = 3.54, *p* = 0.017), the right frontal-central area (MD_HSD_ = 2.48, *p* = 0.03) and the right central area (MD_HSD_ = 1.98, *p* = 0.036), but greater amplitudes in the middle central area (MD_HSD_ = −2.91, *p* = 0.052).

No significant main effect of Group (F < 1) and Group by other important factor interactions were observed: Group by Agreement interaction (F < 1), Group by Agreement by Coronal interaction (F < 1), Group by Coronal interaction (F (4, 84) = 1.96, *p* = 0.13, epsilon _H-F_ = 0.700, η^2^_p_ = 0.085), Group by Anterior–posterior interaction (F (4, 84) = 2.2, *p* = 0.14, epsilon _H-F_ = 0.426, η^2^_p_ = 0.094), and Group by Agreement by Anterior–posterior by Coronal interaction (F (16, 336) = 1.18, *p* = 0.31, epsilon _H-F_ = 0.534, η^2^_p_ = 0.053).

### 3.2. ERP Time Window 500–850 ms

Though there was a significant Group by Agreement by Anterior–posterior interaction (F(4, 84) = 4.95, *p* = 0.018, epsilon _H-F_ = 0.409, η^2^_p_ = 0.191), post hoc analyses did not show a clear effect of Agreement on this positive waveform (i.e., P600) at posterior regions. TD children showed a trend towards an effect of Agreement in the frontal area (MD_HSD_ = 1.58, *p* = 0.068) but not in the central-parietal area (MD_HSD_ = −0.43, *p* = 0.63) or in the parietal area (MD_HSD_ = −1.01, *p* = 0.28). In contrast, children with DLD displayed no effect of Agreement in any of these areas (frontal: MD_HSD_ = −0.51, *p* = 0.95); central-parietal (MD_HSD_ = 0.60, *p* = 0.52); (parietal: MD_HSD_ = 0.31, *p* = 0.75).

There was a significant Group by Coronal interaction (F (4, 84) = 3, *p* = 0.036, epsilon _H-F_ = 0.760, η^2^_p_ = 0.13) and there was also a significant Group by Anterior–posterior by Coronal interaction (F(16, 336) = 3.1, *p* = 0.01, epsilon _H-F_ = 0.335, η^2^_p_ = 0.13). For both match and mismatch conditions, TD children displayed larger positive waves than children with DLD in the right frontal area (MD_HSD_ = 1.98, *p* = 0.08) and in the right frontal-central area (MD_HSD_ = 1.4, *p* = 0.08). This effect was observed in the opposite direction (i.e., TD children displayed a smaller positive wave than children with DLD) in the middle central area (MD_HSD_ = −2.28, *p* = 0.047), middle central-parietal area (MD_HSD_ = −2.62, *p* = 0.021), and middle and middle-right parietal area (MD_HSD_ = −1.99, *p* = 0.077 and MD_HSD_ = −1.7, *p* = 0.065).

There were no significant main effects of Group (F < 1) and Group by other important factor interactions: Group by Agreement interaction (F < 1), Group by Agreement by Coronal interaction (F < 1), Group by Anterior–posterior interaction (F (4, 84) = 1.30, *p* = 0.28, epsilon _H-F_ = 0.434, η^2^_p_ = 0.058), and Group by Agreement by Anterior–posterior by Coronal interaction (F < 1).

## 4. Discussion

To summarize, we have seen that typically developing children show ERP LAN effects that were greater when presented with agreement mismatches than with agreement matches. This was true in the frontal and frontal-central areas. This was not the case for children with DLD, however. Further, there was a near-significant P600 type effect for TD children, but not DLD children, in the frontal region. We believe that the non-significance of the P600 in the TD sample is due to our relatively smaller sample size and to the wide variance in our measurements.

Thus, with regard to our research questions, typically developing children showed significant differences, particularly with regard to the dimension of the ERP thought to correspond to morphosyntax (LAN), between agreement match and agreement mismatch. In contrast, this was not true of children with DLD. This suggests that they are not as sensitive to this type of ungrammaticality, unlike their same-aged, typically developing peers.

Our results are consistent with the large majority of behavioral studies of clitic agreement in Spanish-speaking children with DLD, both monolingual [57,64,65] and bilingual [44,45,66,67,68]. In this way, our study provides converging evidence of the DLD deficit in this domain of grammar, independently of the task demands inherent in the behavioral measures that have been used previously.

With respect to theoretical accounts of the DLD deficit, these results appear consistent with both the Interface Deficit as well as the Computational Complexity Hypothesis (and possibly the Unique Checking Constraint, with the modifications alluded to above).

Conceptually, there are a number of potential causes to which we could attribute our results, and the apparent consensus finding in the literature, that child Spanish-speakers with DLD fail to produce or detect adult-like gender agreement on direct-object clitic pronouns. First, it could be the case that children with DLD have weak, less well-developed lexical representations of the gender of the nouns with which adjectives agree. Such a claim would be supported by the fact that children with DLD have been consistently shown to have smaller vocabularies than typically developing age matches (e.g., [93,94,95]). If this were the only problem, however, we might not expect morphosyntactic problems with areas of grammar that do not depend on lexical development, such as null subjects and determiners, which are problematic for children with DLD (e.g., [48,50]), and which have across-the-board morphosyntactic properties, not typical of lexically dependent processes, as with clitics.

Another possible candidate cause is working memory. There is substantial evidence that children with DLD have fewer working memory resources to work with than do typically developing children (e.g., [96,97,98,99]). Working memory is obviously relevant to pronominal coreference where that coreference is anaphoric in nature. That is, it is one thing to have lexical representations of gender, but it is another to be able to hold these representations in memory so that morphosyntax can process them. One argument against such an explanation, however, comes from work by Noonan et al. [100], who showed that children with DLD may present with or without deficits in working memory. Specifically, they demonstrated that children diagnosed with DLD judge tense errors as ungrammatical less than typically developing controls. However, those who were also diagnosed with short-term memory deficits (diagnosed using a verbal working memory task, and two visuospatial working memory tasks) were significantly worse than those who lacked such a diagnosis on judging tense errors that occurred later in the sentence. From this finding, it seems likely that DLD and short-term memory measures are at least somewhat independent of one another. The lexicon, because tense is not lexically dependent, would not seem to play a substantial role here, either. Further, the results presented in the current report are of clitic-antecedent coreference via deixis, not anaphora. The stimuli in our experiment were visible during the auditory presentation of the sentences. In this way, reference was not established via anaphora, but rather by deictic processes, which one imagines require very little in the way of working memory capacity, though this relationship may not be very well understood.

Finally, there is the previously addressed question of whether DLD could consist of a general failure of morphosyntax. As alluded to earlier, there seems to be evidence that this is false, in the sense that child English-speakers [61] and child Spanish-speakers [43] mark plural on nouns as do typically developing controls. Similarly, child Spanish-speakers are not statistically different from age-matched in controls in noun–adjective agreement [43]. Thus, the pattern appears to be that only those dimensions of morphosyntax that are sensitive to discourse—the semantic class of definites—are problematic for children with DLD, while more local, discourse-independent syntactic relations such as plural marking on nouns and noun–adjective agreement are not problematic.

In sum, morphosyntactic constructions are difficult for children with DLD when the constructions depend critically on lexical development, as in the clitic pronouns in our current study, but also when the constructions occur independently of lexical features, as in definite determiners and null subjects. Further, children with DLD appear to have problems with morphosyntactic constructions when there is relatively little working memory load, as in our current study with deictic clitic pronouns, but also when the construction is presented with varying levels of memory load, as in tense marking in Noonan et al. [100]. Finally, it does not seem to be a general problem of morphosyntax itself, but rather the subset of morphosyntactic constructions, which fall in the semantic natural class of definites, that are most problematic.

If an Interface Deficit account is on the right track, then the problem centers on the inability of definites, including direct-object clitics, to link with antecedents in the Conversational Common Ground. On the basis of our current results, it would appear that the Conversational Common Ground is accessed not only via anaphoric processes that likely depend on the possibly domain-general, performance system of working memory, but also on the more direct, possibly domain-specific, linguistic mechanism of deixis. In this way, while there are likely multiple dimensions to the DLD deficit, one dimension characteristic of it may be a general ability to use definites, whether their presupposition of uniqueness (e.g., [101,102]) is satisfied via anaphoric processes or whether it is derived via deixis. In future work, we hope to add greater specificity and empirical substantiation to this speculative claim, with the objective always of giving providing greater understanding of the nature of this disorder.

## Figures and Tables

**Figure 1 children-08-00175-f001:**
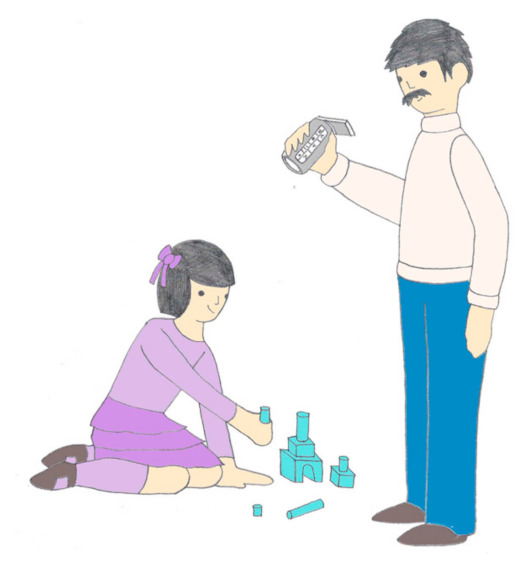
Image to Elicit Gender Agreement Mismatch with the Sentence “The father filmed him.”

**Figure 2 children-08-00175-f002:**
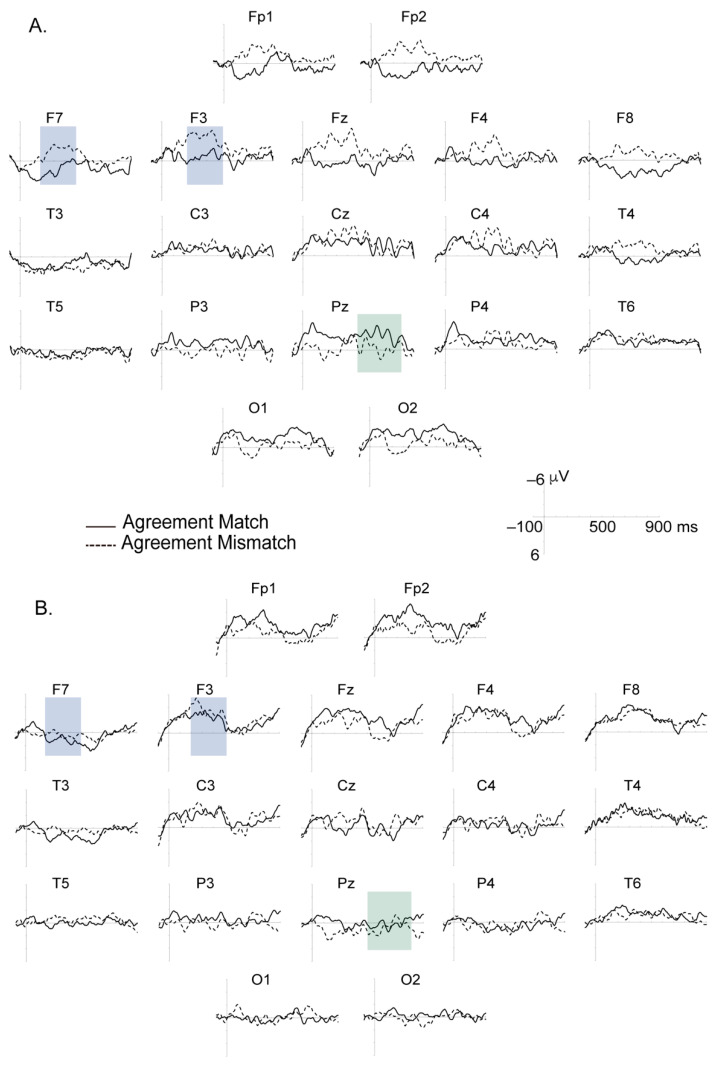
Grand average ERPs of (**A**), the TD children and (**B**), the children with DLD. The gender-mismatch of the clitic in the sentence/picture pair condition (dotted line) is plotted against the gender-match condition (solid line). The axis of the ordinates indicates the onset of the clitic. Negative voltage is plotted up. Purple and green shadow boxes show the time windows analyzed.

**Table 1 children-08-00175-t001:** Summary of Findings from Previous Studies on Clitic-Antecedent Agreement Errors for Spanish-speaking Children with DLD.

Study	Age (Years Old)	Method	% Agreement Errors	Different from Controls?
Monolingual Spanish				
Merino [64]	5–8	Elicited Production	25%	Yes
De la Mora [57]	5.3	Elicited Production–Prompted Response	17%	Yes
Morgan et al. [65]	5.3	Cloze Test	11%	Yes ^a^
Jackson-Maldonado and Maldonado [3]	7.3	Frog Story–Story Retell	Mean number of errors = 1.58	No
Bilingual Spanish-English				
Bedore and Leonard [44,66]	3;11–5;6	Elicited Production	38% ^b^	Yes
Bedore and Leonard [45]	3;11–5;6	Spontaneous Production	8% ^c^	Yes
Jacobson and Schwartz [67]	4;7	Elicited Production	63% feminine 12% masculine	Yes
Jacobson [68]	Lower grades—7.2 years old Higher grades—10.9 years old	Elicited Production	48% ^d^ 46% ^e^ 33% ^f^ 32% ^g^	Yes Yes Yes No
Bilingual Spanish-Catalan				
Bosch and Serra [69]	7;6	Spontaneous Production	18.25%	No

^a^ Significance was for overall clitic differences including substitution and omission, not substitution, by itself. ^b^ Compiled from Bedore and Leonard [66], p. 915, Table 6. ^c^ Compiled from Bedore and Leonard [45], p. 216, Table 8. ^d^ Lower age group, with preverbal clitics, for gender. ^e^ Lower age group, with preverbal clitics, for number. ^f^ Higher age group, with preverbal clitics, for gender. ^g^ Higher age group, with preverbal clitics, for number.

## Data Availability

The data presented in this study are openly available in FigShare on at doi: 10.6084/m9.figshare.13637642. (accessed on 25.01.2021)

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
