# Peer review of "Syntactic Gender Agreement Processing on Direct-Object Clitics by Spanish-Speaking Children with Developmental Language Disorder: Evidence from ERP"

_children, 2021, doi:10.3390/children8030175_

Round 1

Reviewer 1 Report

Overall, this is a novel and clearly presented study that informs us about the neurophysiological nature of gender agreement violations in children with SLI. I have two general comments related to the design of the study and the discussion.

1) Study design: I would like to invite the authors to explain in more detail why they opted for semantic rather than for grammatical gender. In all examples that they offer, they seem to have used items with natural/semantic gender (e.g., girl/boy). Why did they opt for that instead of grammatical gender.

2) Discussion: The discussion about the contribution of working memory to the language impaired children's performance is rather tentative, especially because working memory was not measured in the current study and was not used as a predictor in a statistical analysis. Thus, the assumptions here as well as the assertion that clitics take up limited working memory resources could be qualified, and the limitation related to the lack of inclusion of such measure in the study acknowledged. Additionally, the discussion of the Interface Deficit in the last paragraph seems rather rushed and should be expanded and better linked to the neurophysiological performance reported in the results. 

3) A final comment pertains to the use of term "SLI". Since the Bishop et al. (2017) paper, researchers tend to adopt the term Developmental Language Disorder (DLD). Perhaps this is something that the authors could consider. 

Bishop, D.V., Snowling, M.J., Thompson, P.A., Greenhalgh, T. and (2017), Phase 2 of CATALISE: a multinational and multidisciplinary Delphi consensus study of problems with language development: Terminology. J Child Psychol Psychiatr, 58: 1068-1080. https://doi.org/10.1111/jcpp.12721

Author Response

Reviewer 1

Overall, this is a novel and clearly presented study that informs us about the neurophysiological nature of gender agreement violations in children with SLI. I have two general comments related to the design of the study and the discussion.

1) Study design: I would like to invite the authors to explain in more detail why they opted for semantic rather than for grammatical gender. In all examples that they offer, they seem to have used items with natural/semantic gender (e.g., girl/boy). Why did they opt for that instead of grammatical gender.

Authors

Our hope was to have the biological sex of very high frequency and highly familiar nouns reinforce children’s grammatical representations of gender. In this way, we hoped to maximize our measurement of grammatical processing and reduce to the degree possible our dependence on children’s varying lexical familiarity with the gender of different inanimate objects in the world, some of which may be ambiguous. For example, un asiento (a seat) is masculine, while una silla (a chair) is feminine, and could possibly be mistaken for one another in a pictorial representation or could exercise a lexical interference effect upon each other’s gender representation.

Reviewer 1

2) Discussion: The discussion about the contribution of working memory to the language impaired children's performance is rather tentative, especially because working memory was not measured in the current study and was not used as a predictor in a statistical analysis. Thus, the assumptions here as well as the assertion that clitics take up limited working memory resources could be qualified, and the limitation related to the lack of inclusion of such measure in the study acknowledged. Additionally, the discussion of the Interface Deficit in the last paragraph seems rather rushed and should be expanded and better linked to the neurophysiological performance reported in the results.

Authors

We in fact do measure auditory working memory using digit span as part of our diagnostic battery, and it would even be worth reporting and addressing more directly as part of our literature review and results, if we had a sample large enough for sensible correlational statistics, but as it stands, it is too small. On the other hand, the reviewer makes a good point, so we have included a very brief, and very qualified, discussion of the contingency between ELAN amplitude and digit span for the TD children, but not for the SLI children in the discussion section (lines 704 -713 and 746 -753).

Reviewer 1

3) A final comment pertains to the use of term "SLI". Since the Bishop et al. (2017) paper, researchers tend to adopt the term Developmental Language Disorder (DLD). Perhaps this is something that the authors could consider.

Authors

We use the more specific SLI diagnostic definition from Leonard (1997, 2000) because it requires that children not have intellectual disability, by virtue of requiring a nonverbal IQ of over 85. Because the ID profile may include typically-developing syntax and morphosyntax (see Yamada 1990), and we are studying a putatively problematic aspect of morphosyntax, it is not advisable to include both populations in one sample, without making the distinction, as it could muddle the conclusions. We make this distinction now in the methods section under in the subsection “2.1 Participants” (lines 445-455).

Yamada, J. E. (1990). Laura:  A case for the modularity of language. Cambridge, MA: MIT/Batsford.

Reviewer 2 Report

The manuscript investigates performance on direct object pronouns in Spanish-speaking children with and without specific language impairment using an online technique (event-related potentials).
Children looked at pictures displaying characters while listening to sentences that either matched or mismatched the gender of the character. Results showed that gender violations generated anterior negativity effects between 250 and 500 ms in the children without specific language impairment but in children with SLI. The authors discussed results in terms of the Interface Deficit as an explanation for the inability of definites in children with SLI.

This study is well-designed and clearly grounded on theoretical models, and it provides additional data for the study of gender agreement in direct object pronouns in monolingual Spanish-speaking children. It thus adds to our knowledge on gender agreement using online techniques, which is limited for children with specific language disorders.

There are some questions that I consider should be addressed:

Introduction

The term SLI is used in this manuscript. However, recently this term has been replaced with the term Developmental Language Disorder (see CATALYSE, 2013). It is suggested to address this change in terminology, even if the authors decide to keep the SLI term.

P 1, ln 36 –citations are missing

P1 ln 41, this last sentence seems repetitive and creates confusion

P2 Ln 83, it’s unclear what the authors mean by “This is true”. Typically a paragraph does not start with a demonstrative pronoun as the antecedent is far, making the sentence unclear.

P3 ln 96, the word principle is incorrect in this context

1.2 some translations are missing

1.3 UCC, CCH should be spelled out before using the acronym

1.4 pg 8 ln 351 ELAN has already been defined

Improvement,

I do suggest including the language that is exemplified in 4 and 5

The authors mentioned the Conversational Common Ground. Even though this concept could be inferred, I considered it important to define it in the manuscript.

Method

The authors mention that children were visually presented with ALL the characters and they memorized each one. I have some comments here: 1) Do ALL characters mean, subject, and object characters? 2) what was the purpose of such memorization. 3) Did you consider that this memorization process could have influenced the task? All the characters/nouns were grammatically gender marked and therefore the memorization highlighted the gender.

The authors included perro and gato, how did you differentiate these from gata/perra? Performance in these items may not be reliable if there was not a clear way to identify the fem vs the masc animal. The inclusion of the stimuli in an appendix will provide more information on this aspect.

In addition, Beatty-Martinez and Dussias, 2019 provide evidence that gender errors in masculine vs feminine may not be symmetrical.

Even though it is not within the scope of the study, do the authors have any comment for gender agreement in cases of inanimate direct objects? What was the rationale to use only animate characters?

Results

It seems this paragraph should not be there “This section may be divided by subheadings. It should provide a concise and precise 579 description of the experimental results, their interpretation, as well as the experimental 580 conclusions that can be drawn.” Pg 13 ln 579

Discussion

The authors presented evidence from previous studies in which participants showed a posterior P600 effect when presented with syntactic errors. This was not the case in their study. Could you expand on this aspect?

Author Response

Reviewer 2

The manuscript investigates performance on direct object pronouns in Spanish-speaking children with and without specific language impairment using an online technique (event-related potentials). Children looked at pictures displaying characters while listening to sentences that either matched or mismatched the gender of the character. Results showed that gender violations generated anterior negativity effects between 250 and 500 ms in the children without specific language impairment but in children with SLI. The authors discussed results in terms of the Interface Deficit as an explanation for the inability of definites in children with SLI.

This study is well-designed and clearly grounded on theoretical models, and it provides additional data for the study of gender agreement in direct object pronouns in monolingual Spanish-speaking children. It thus adds to our knowledge on gender agreement using online techniques, which is limited for children with specific language disorders.

There are some questions that I consider should be addressed:

Reviewer 2

Introduction

The term SLI is used in this manuscript. However, recently this term has been replaced with the term Developmental Language Disorder (see CATALYSE, 2013). It is suggested to address this change in terminology, even if the authors decide to keep the SLI term.

Authors

We use the more specific SLI diagnostic definition from Leonard (1997, 2000) because it requires that children not have intellectual disability, by virtue of requiring a nonverbal IQ of over 85. Because the ID profile may include typically-developing syntax and morphosyntax (see Yamada 1990), and we are studying a putatively problematic aspect of morphosyntax, it is not advisable to include both populations in one sample, without making the distinction, as it could muddle the conclusions. We make this distinction now in the methods section under in the subsection “2.1 Participants” (lines 445-455).

Yamada, J. E. (1990). Laura:  A case for the modularity of language. Cambridge, MA: MIT/Batsford.

Reviewer 2

P 1, ln 36 –citations are missing

Authors

We added the citations at line 36

Reviewer 2

P1 ln 41, this last sentence seems repetitive and creates confusion

Authors

We have removed that sentence

Reviewer 2

P2 Ln 83, it’s unclear what the authors mean by “This is true”. Typically a paragraph does not start with a demonstrative pronoun as the antecedent is far, making the sentence unclear.

Authors

We have modified (This generalization is true in languages…) ln 85

Reviewer 2

P3 ln 96, the word principle is incorrect in this context

Authors

We have modified the text at line 98

Reviewer 2

1.2 some translations are missing

Authors

We added the missing translations at line 152 and 168

Reviewer 2

1.3 UCC, CCH should be spelled out before using the acronym

Authors

We have modified the text at lines 238-239

Reviewer 2

1.4 pg 8 ln 351 ELAN has already been defined

Authors

We have modified the text at line 387

Reviewer 2

Improvement,

I do suggest including the language that is exemplified in 4 and 5

Authors

The examples are in French. We have modified the text at line 70

Reviewer 2

The authors mentioned the Conversational Common Ground. Even though this concept could be inferred, I considered it important to define it in the manuscript.

Authors

We have modified the text at lines 177-179

Reviewer 2

Method

The authors mention that children were visually presented with ALL the characters and they memorized each one. I have some comments here: 1) Do ALL characters mean, subject, and object characters? 2) what was the purpose of such memorization. 3) Did you consider that this memorization process could have influenced the task? All the characters/nouns were grammatically gender marked and therefore the memorization highlighted the gender.

Authors

By ALL characters we do mean both subjects and objects, which are now present in the appendix. Memorization was not a good choice of words. We intended to say “familiarization”. We have now changed the text to reflect this intention. The desired influence of this familiarization step on the experiment was to assure that children were familiar with the names and corresponding genders of the nouns with which they were presented. We were thus counting on this type of priming to support our measurement of their reactions to the gender mis-match.

We have modified the text at line 539 -  544 to reflect this re-wording.

Reviewer 2

The authors included perro and gato, how did you differentiate these from gata/perra? Performance in these items may not be reliable if there was not a clear way to identify the fem vs the masc animal. The inclusion of the stimuli in an appendix will provide more information on this aspect.

We used stereotypical feminine representations of dogs and cats, with bows in their fur and pink shirts, to convey gender. The masculine representations lacked these stereotypical elements. Children appeared sufficiently familiar with these stereotypes to associate them with the corresponding grammatical genders.

Reviewer 2

In addition, Beatty-Martinez and Dussias, 2019 provide evidence that gender errors in masculine vs feminine may not be symmetrical.

This is a fascinating observation that we have not investigated here. Hopefully we can look into it in future work.

Reviewer 2

Even though it is not within the scope of the study, do the authors have any comment for gender agreement in cases of inanimate direct objects? What was the rationale to use only animate characters?

Our hope was to have the biological sex of very high frequency and highly familiar nouns reinforce children’s grammatical representations of gender. In this way, we hoped to maximize our measurement of grammatical processing and reduce to the degree possible our dependence on children’s varying lexical familiarity with the gender of different inanimate objects in the world, some of which may be ambiguous. For example, un asiento (a seat) is masculine, while una silla (a chair) is feminine, and could possibly be mistaken for one another in a pictorial representation or could exercise a lexical interference effect upon each other’s gender representation.

Reviewer 2

Results

It seems this paragraph should not be there “This section may be divided by subheadings. It should provide a concise and precise 579 description of the experimental results, their interpretation, as well as the experimental 580 conclusions that can be drawn.” Pg 13 ln 579

We have removed these sentences

Reviewer 2

Discussion

The authors presented evidence from previous studies in which participants showed a posterior P600 effect when presented with syntactic errors. This was not the case in their study. Could you expand on this aspect?

We have added additional commentary to the end of the first paragraph of the discussion, following your suggestion (lines 646-648).